# Virtual Training System for the Teaching-Learning Process in the Area of Industrial Robotics

Jordan S. Ipiales *, Edison J. Araque, Víctor H. Andaluz * and César A. Naranjo

Departamento de Eléctrica y Electrónica, Universidad de las Fuerzas Armadas ESPE, Sangolquí 171103, Ecuador
* Correspondence: jsipiales@espe.edu.ec (J.S.I.); vhandaluz@espe.edu.ec (V.H.A.); Tel.: +593-958-779-578 (V.H.A.)

**Abstract:** This paper focuses on the development of a virtual training system by applying simulation techniques such as: Full Simulation and Hardware-in-the-Loop (HIL). This virtual reality system is intended to be a teaching and learning tool focused on the area of industrial robotics. For this purpose, mathematical models (kinematic and dynamic) have been considered. These models determine the characteristics and restrictions of the movements of a Scara SR-800 robot. The robot is then virtualized to simulate position and trajectory tasks within virtual environments. The Unity 3D graphic engine (Unity Software Inc., San Francisco, CA, USA), allows the design and development of the training system which is composed of a laboratory environment and an industrial environment. The same that contribute to the visualization and evaluation of the movements of the robot through the proposed control algorithm using the mathematical software (MatLab, manufactured by MathWorks, USA), through shared memories. This software in turn can be linked to an electronic board (Raspberry Pi) for data acquisition through a wireless connection. Finally, the stability and robustness of the implemented controllers are analyzed, as well as the correct operation of the virtual training system.

**Keywords:** learning; scara robot; hardware-in-the-loop; stability; robustness

## 1. Introduction

The synchronization of tasks with manufacturing equipment, manipulation of work pieces and use of devices are based on the application of industrial automation [1], This is how the fourth revolution appears, where production systems become changing, thus obtaining more efficient and higher quality processes. So it is important technologies within the industrial processes contributing to the connectivity of the virtual world where access to services in the Internet of things becomes essential; this leads to the development of cyber-physical systems, big data, digital information exchange and intelligent robots which are called notable and characteristic principles within Industry 4.0 and IoT as exposes [2,3].

Where the growth of robotics worldwide has been part of it, i.e., it has had a great impact on the industrial sector in the intelligent automation applied by developed countries [4]. Asia is the world's largest industrial robot market with a compound annual growth rate (CAGR) of 18%. China, Japan, the United States, the Republic of Korea, and Germany account for 78% of the robot installations in the world. For this reason, an analysis by the International Federation of Robotics (IFR) determined a worldwide average robot density in the manufacturing industry of approximately 141 robots per 10,000 employees. On the other hand, in reference to industrial sectors such as the electrical and electronic sector became the main customer for the installation of industrial robots with 26% in 2021 surpassing the automotive sector which has 23% of robots implanted in factories, followed by the metal and machinery industry with 12%; the chemical sector with 5% and the food and beverage industry with 3% [5].

In 2021 the operational stock of industrial theft reaches 3,477,127 installed units worldwide [5]. The commercial brands that manufacture industrial manipulator robots are ABB, Fanuc, Kuka, Yaskawa, Stäubli, Kawasaki, and Epson [6]. The IFR Robot Suppliers Committee classifies industrial manipulator robots according to the mechanical structure of the

arm as follows: Cartesian robots corresponding to linear and gantry robots; Scara robots; articulated robots, which are anthropomorphic robots; parallel robots, which are delta robots; cylindrical robots; and among others [5]. (i) Cartesian robot: this type of robot has a limited working space, shows a serial configuration of the motors so it performs a linear movement, and cannot be subjected to high accelerations because there is an error in the position of the axis, reducing structural rigidity [7], are also used in specific applications for the handling of heavy loads [6]. (ii) Delta robot: they are mostly used for tasks of handling, sorting and classification of light materials where the development of this type of robots with high velocity and precision and light weight means that the links are made of flexible materials which cause vibrations as mentioned in the following table [8,9]. (iii) Anthropomorphic Robot: it has a very versatile configuration giving way to its movements having a similarity with the human arm, however the large amount of data set causes an erroneous prediction and at the same time there is an excessive consumption of energy of the links [10,11]. (iv) Scara Robot: this type of manipulator is widely used in production processes where repetitive tasks with high velocity and precision are required considering the manipulation of small loads [6]; the most relevant ones are: assembly, welding in the automotive area, adhesive placement, screwing, material distribution, part inspection and object transfer, which are present in sectors such as: pharmaceutical, aerospace, food, etc., i.e., in all automated systems [12,13].

Therefore, as described above, there is great interest in both the scientific community and the industrial sector in developing new control techniques, so that robots can work cooperatively with humans and thus perform tasks together within an environment. Therefore, to obtain the physical structure of this type of robotic systems as well as its commissioning is impossible, being the cost its main problem and within the maintenance of robots downtime directly affects the waste of manpower, testing and production [14], furthermore, the safety and ergonomics for the operator in the performance of cyber-physical activities, as explained by [15,16].

Through Virtual Reality (VR), which is based on 3D models applied to new products and processes, it has greater relevance in the implementation of Industry 4.0 [17], leveraging powerful and flexible tools such as closed-source software; due to robot manufacturing brands, robots have limitations, which directly affect the implementation of new control algorithms. On the other hand, there is open source software that generates lower costs and more innovative results [18] so they have great accessibility for the development of virtual environments using graphic engines such as: Unity 3D, Unreal, Godot Engine and Cryengine, etc [19,20], where a wide range of scenarios, objects with realistic dynamics, and intelligent characters are developed. The software that facilitates the simulation of the Scara robot as well as the environments is the Unity 3D graphics engine, which contributes to the implementation of the Hardware-in-the-Loop technique [21].

Consequently, this article presents an interactive and immersive virtual training system focused on teaching and learning. This virtual system consists of a laboratory environment and an industrial environment which simulate the behavior of the Scara SR-800 robot using the Full Simulation technique and the HIL technique. A kinematic model and a dynamic model were obtained with velocities instead of torques as input signals. In addition, these mathematical models allow the design of the proposed control scheme. Through the use of software (MatLab, manufactured by MathWorks, USA), the advanced control algorithm is implemented and communication with the Unity 3D software is established in real time. This is done through the shared memories developed by the authors. Due to the lack of a real robotic system because of its high cost and danger, the Hardware-in-the-Loop technique is used. It consists in the exchange of data between a target software (MatLab, manufactured by MathWorks, USA) and a Raspberry Pi 4 board (manufactured by the Raspberry PI Foundation, UK) that will simulate the real behavior of the Scara robot. All this using the ZIGBEE communication protocol. Finally, to evaluate the acceptance of the virtual training system, the results of a usability test are presented.

The following document consists of seven sections. Section 2 details the methodology used for the development of the virtual training system. Section 3 represents the virtualization structure of the environments applying Full Simulation by means of shared memories and the HIL technique that consists in the connection of the Raspberry Pi 4 electronic card with the computer. Section 4 describes the modeling of the Scara manipulator robot and includes mathematical modeling. Section 5 exposes a kinematic controller and a dynamically compensated controller. Section 6 presents the results of the interactive virtual environments and, in addition, the percentage of usability. Section 7 shows the conclusions obtained from the performance of the interactive virtual environments.

## 2. Methodology

Figure 1 presents the development of the Hardware–in–the–Loop (HIL) technique. This is a simulation technique that allows to visualize the behavior of complex systems, developed in real time. This HIL is performed in the absence of a real robotic system due to its high cost and danger. For this reason, an electronic card is used that will contain the mathematical modeling of the robot and will establish a communication with a target software. This allows emulating the behavior of the Scara SR-800 robot through the implementation of control algorithms.

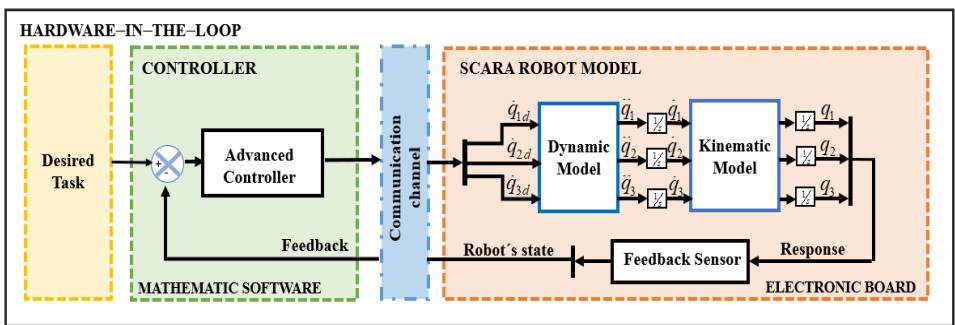

**Figure 1.** Hardware–in–the–Loop (HIL) block diagram.

Moreover, it is presented: the structure of the methodology for the design of the virtual training system. It is focused on the teaching and learning process. Figure 2 shows the stages to implement the 3D virtual simulator.

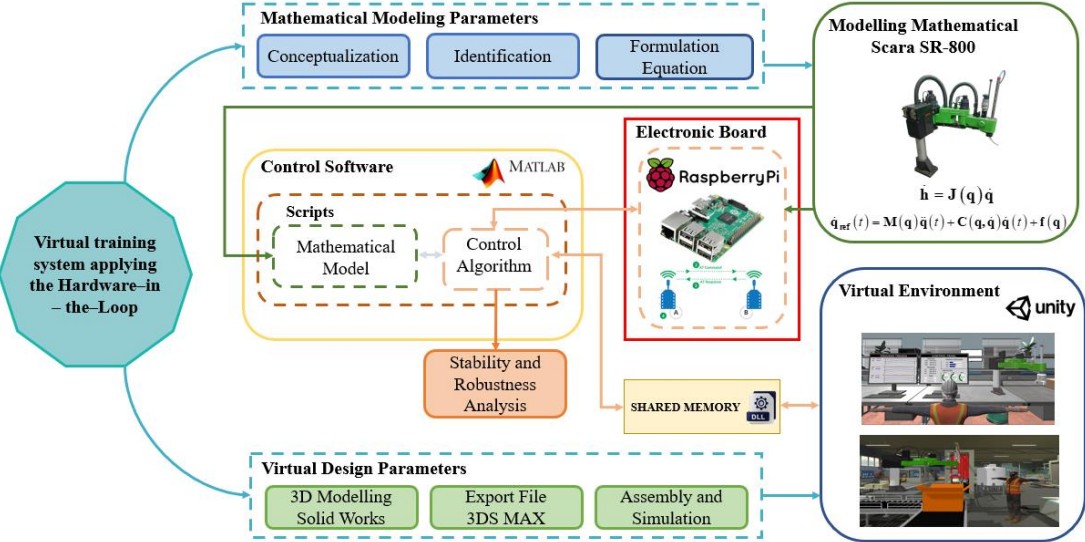

**Figure 2.** Methodology for the control and virtualization of the Scara SR-800 Robot.

The methodology developed consists of the following stages: (i) Mathematical Modeling, which using modeling parameters such as: conceptualization, identification and formulation of equations, a kinematic model and a dynamic model of a real robot are obtained, which are represented in a compact form. (ii) Virtualization of Environments, in this stage the 3D design of the virtual environments and the robotic system is carried out using CAD tools, as well as the elements that constitute the design. (iii) Controller Design, the proposed control algorithm allows to evaluate the behavior of the robotic system when performing autonomous position and trajectory tasks. The stability and robustness of the controller for the virtual training system is analyzed [22]. In addition, the Full simulation technique uses Unity3D software and MatLab software for data exchange via shared memories. Finally, to apply the HIL technique, an electronic board is used in conjunction with MatLab and Unity3D software.

## 3. Virtualization–Full Simulation–HIL

In this section we present the virtualization of the virtual training system as shown in Figure 3. Applying the technique (a) Full Simulation, which consists in the use of the Unity3D software that contains the virtual training system. While the MatLab software has in its programming scripts the mathematical model of the robot and the proposed control algorithm, the MatLab software has in its programming scripts the mathematical model of the robot and the proposed control algorithm. For data exchange between these two softwares, a bilateral communication is carried out through shared memories; (b) To apply the HIL technique, an electronic card is used. The mathematical model of the real robot, both kinematic and dynamic, is entered through programming lines and a wireless communication channel is established using the ZIGBEE protocol. All this contributes to the exchange of data between the Raspberry Pi 4 and MatLab, the latter owns the control algorithm and interacts with Unity through the same shared memories described above.

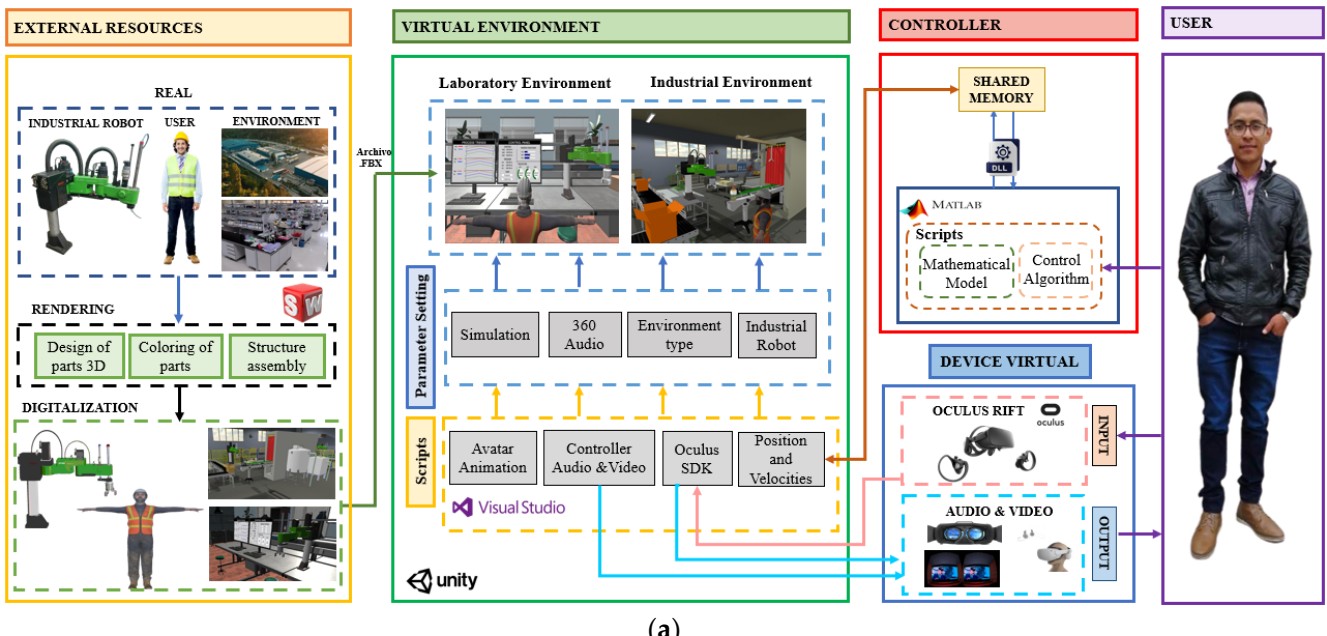

(**a**)

**Figure 3.** *Cont.*

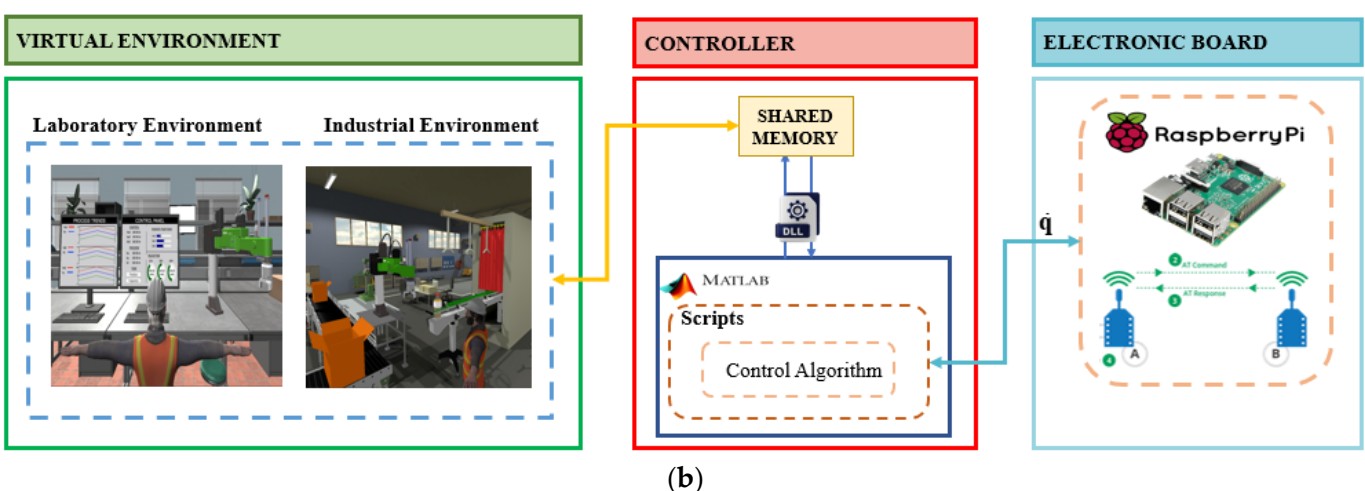

(**b**)

**Figure 3.** Virtualization scheme of the environments and the Scara SR-800 robot for the training system using the Full Simulation technique and the HIL technique. (**a**) Training system applying the Full simulation technique through a virtual environment. (**b**) Training system using the Hardware–in–the–Loop (HIL) technique through a virtual environment.

For the virtualization of the system, have as external resources the laboratory environment, the industrial environment, as well as the user and the SR-800 industrial robot. The design of this robot was carried out by means of 3D CAD design in SolidWorks software. Where a file was obtained in ". STL" file was obtained and converted to a ".fbx" file by means of 3DS Max software. Finally, these parts were exported and assembled in Unity 3D software, as shown in Figure 4.

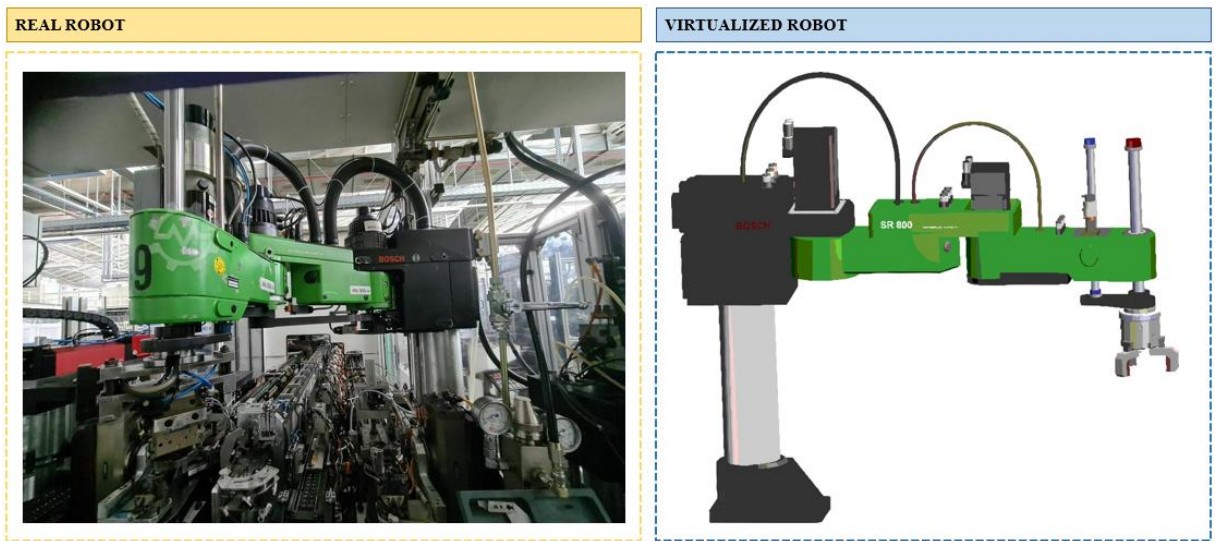

**Figure 4.** Virtualization of the Scara SR-800 robot.

In addition, the Unity 3D graphics engine allows the development of the environments that conform the virtual training system, such as the laboratory environment and the industrial environment, which were digitized in ".fbx" format. And they are focused on the area of industrial robotics that will be useful for the teaching and learning process. It is worth mentioning that these environments will facilitate the interaction of the robot's movements when performing position and trajectory tasks assigned by the user.

Programming scripts are a fundamental part of the development of virtual environments. They contain lines of code that emulate the movement of the Scara robot. These movements are generated by means of mathematical modeling, both kinematic and dy-

namic, previously obtained. Some scripts allow the management of libraries (SDK), which contribute to the communication of virtual input and output devices, improving the interaction with the developed environments. As well as for the control of audio, video, disturbances, animation of the avatar and other objects new scripts are added.

For the implementation of shared memory (SM), a dynamic link library (dll) [23], which is instantiated at the time of starting the virtual training system [24], which generates a shared memory in RAM (SM) for the exchange of information in real time between the MatLab-Unity3D software. The dll provides permissions for: (i) tagging memory space; (ii) functions such as modifying/getting stored information; and (iii) freeing memory space [25]. Figure 5 shows the information exchange scheme between MatLab-Unity3D. In the first instance the memory spaces are created from the MatLab, which send the corresponding data from robot A and robot B as: desired positions of $\mathbf{h_{dA}} = \begin{bmatrix} h_{xd}, h_{yd}, h_{zd} \end{bmatrix} \in \mathrm{R}^3$, $\mathbf{h_{dB}} = \begin{bmatrix} h_{xd}, h_{yd}, h_{zd} \end{bmatrix} \in \mathrm{R}^3$. Real positions $\mathbf{h_A} = [h_x, h_y, h_z] \in \mathrm{R}^3$, $\mathbf{h_B} = [h_x, h_y, h_z] \in \mathrm{R}^3$. Robot joint positions $\mathbf{q_A}(q_1, q_2, q_3)$, $\mathbf{q_B}(q_1, q_2, q_3)$. Angular velocities of robots $\dot{\mathbf{q}}_\mathbf{A}(\dot{q}_1, \dot{q}_2, \dot{q}_3)$, $\dot{\mathbf{q}}_\mathbf{B}(\dot{q}_1, \dot{q}_2, \dot{q}_3)$; while Unity3D sends the states of the process $\chi(c, p, b, r, v, u)$ such as: change of position $c$, change of trajectory $p$, disturbance activation $b$, bottle detector $r$, conveyor belt activation $v$, failure indicator $u$, thus closing the control loop.

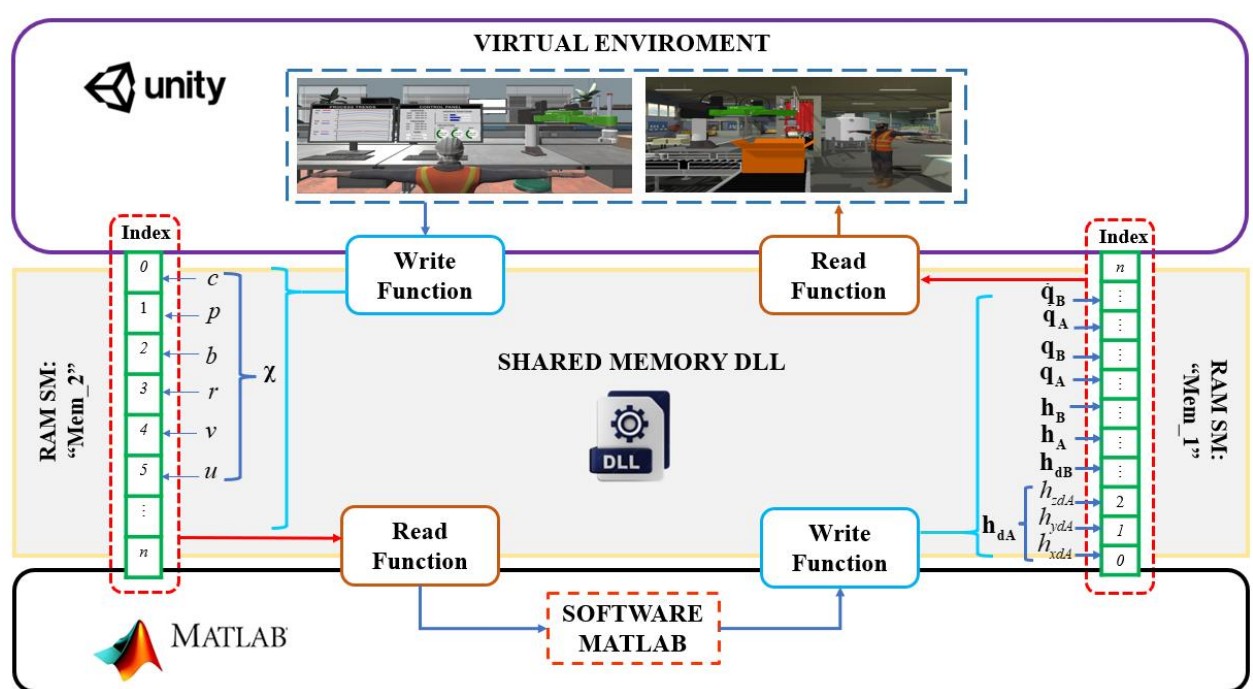

**Figure 5.** Information exchange structure between MatLab and Unity3D through shared memories.

## 4. Modeling

### 4.1. Kinematics of the Robot Manipulator Scara Bosh

The Scara robot is one of the most widely used in the industrial sector due to its versatility when performing repetitive and autonomous tasks at high velocities.

Figure 6 shows the configuration of the manipulator robot to be modeled; where $\mathbf{h} \in \mathbb{R}^m$ with $m = 3$ represents the location of the operating end with respect to the inertial reference frame $\{R\}$ as follows:

$$\mathbf{h} = \begin{bmatrix} h_x \\ h_y \\ h_z \end{bmatrix} = \begin{bmatrix} l_1 \cos(q_1) + l_2 \cos(q_1 + q_2) \\ l_1 \sin(q_1) + l_2 \sin(q_1 + q_2) \\ d - q_3 \end{bmatrix} \tag{1}$$

where $d$, $l_1$, $l_2$ are dimensions of the robotic arm; $q_1$ and $q_2$ represents the rotational motion of each link, while $q_3$ represents the linear displacement on the axis $Z$.

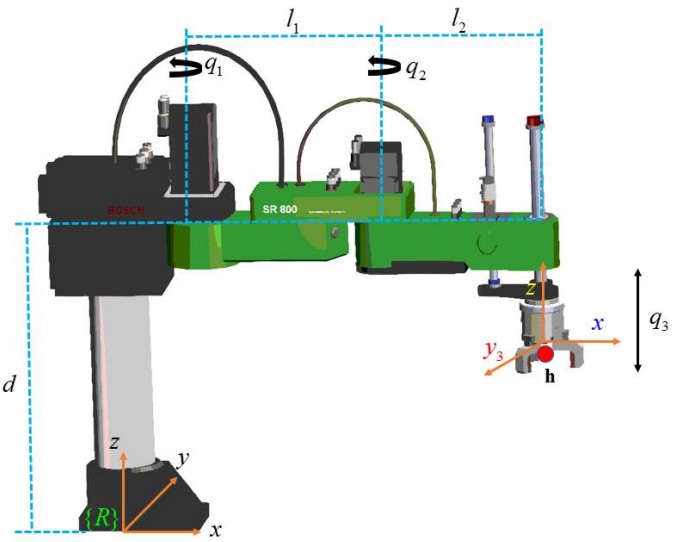

**Figure 6.** Scara SR-800 Industrial Robot.

Therefore, by deriving Equation (1) is obtained the kinematic model of the manipulator robot, where $\dot{\mathbf{h}}$ represents the operating end velocities at the coordinates of $\{R\}$, $y$ $\dot{\mathbf{q}}$ represents the velocities of each joint. The kinematic model is defined as:

$$
\begin{bmatrix} \dot{h}_x \\ \dot{h}_y \\ \dot{h}_z \end{bmatrix} = \begin{bmatrix} -l_1\sin(q_1) - l_2\sin(q_1+q_2) & -l_2\sin(q_1+q_2) & 0 \\ l_1\cos(q_1) + l_2\cos(q_1+q_2) & l_2\cos(q_1+q_2) & 0 \\ 0 & 0 & -1 \end{bmatrix} \begin{bmatrix} \dot{q}_1 \\ \dot{q}_2 \\ \dot{q}_3 \end{bmatrix}
\tag{2}
$$

$$
\dot{\mathbf{h}} = \mathbf{J}(\mathbf{q})\dot{\mathbf{q}}
\tag{3}
$$

where $\mathbf{J}(\mathbf{q}) \in R^{m \times n}$ with $m = n = 3$ represents the Jacobian matrix of the manipulator robot; $\mathbf{q} = \begin{bmatrix} q_1 & q_2 & q_3 \end{bmatrix}$ where $q_1$ and $q_2$ represent rotational joints and $q_3$ is a prismatic joint.

### 4.2. Dynamic Model of the Scara Manipulator Robot

The dynamic model represents the dynamics of the robotic system, which can be obtained by using the Euler-Lagrange dynamic equations, which are based on the conservation of energy (kinetic and potential). The same ones that allow to find the torques produced by the rotational joints, for this work we consider the dynamic model represented in [26].

$$
\boldsymbol{\tau_P} = \overline{\mathbf{M}}(\mathbf{q})\ddot{\mathbf{q}} + \overline{\mathbf{C}}(\mathbf{q},\dot{\mathbf{q}})\dot{\mathbf{q}} + \overline{\mathbf{f}}(\mathbf{q})
\tag{4}
$$

where $\mathbf{q} \in \mathbb{R}^n$ is the position vector; $\overline{\mathbf{M}}(\mathbf{q}) \in \mathbb{R}^{n \times n}$ is the inertia matrix; $\overline{\mathbf{C}}(\mathbf{q},\dot{\mathbf{q}}) \in \mathbb{R}^{n \times n}$ is the matrix of centripetal and centrifugal forces; $\overline{\mathbf{f}}(\mathbf{q}) \in \mathbb{R}^n$ is the vector of frictional forces. Forces generated by the movement of individual links, and $\boldsymbol{\tau_P} \in \mathbb{R}^n$ is the vector of input torques.

The mathematical model of two DC motors is used to determine the necessary torques used by the two links of the robotic system in general, at the time of performing position and trajectory tasks in the different scenarios, given by:

$$
\tau_d = \frac{k_a\left(v_i - k_b\dot{q}_1\right)}{R_c}
\tag{5}
$$

$$\tau_l = \frac{k_e(v_j - k_w \dot{q}_2)}{R_y} \tag{6}$$

where $\tau_d$, $\tau_l$ represent the torques for the movement of the links of the robotic system; $v_i$, $v_j$ are the supply voltages; $\dot{q}_1$, $\dot{q}_2$ are the angular velocities; $k_a$, $k_e$ is the result of the product of the torque constant and the reduction constant; $k_b$, $k_w$ is also the result of the multiplication between the counter electromotive constant and the reduction constant; finally $R_c$, $R_y$ are the electrical resistances of motor 1 and motor 2 respectively. Regrouping the terms of Equations (5) and (6) to obtain a compact structure for the controller design can be expressed as follows:

$$\begin{bmatrix} \tau_d \\ \tau_l \end{bmatrix} = \begin{bmatrix} \frac{k_a}{R_c} & 0 \\ 0 & \frac{k_e}{R_y} \end{bmatrix} \begin{bmatrix} v_i \\ v_j \end{bmatrix} - \begin{bmatrix} \frac{k_a k_b}{R_c} & 0 \\ 0 & \frac{k_e k_w}{R_y} \end{bmatrix} \begin{bmatrix} \dot{q}_1 \\ \dot{q}_2 \end{bmatrix} \tag{7}$$

$$\boldsymbol{\tau_p} = \mathbf{S}\mathbf{v_r} - \mathbf{G}\dot{\mathbf{q}} \tag{8}$$

where $\mathbf{v_r}$ is the vector of supply voltages for each motor of the robotic system; and $\mathbf{S}$, $\mathbf{G}$ are gain matrices.

The power supply voltages for the Scara robot can be described in terms of a PD controller, very commonly used in industrial robots. This type of controller assigns the power supply voltage to the motor represented in:

$$v_i = k_{P1}\left(\dot{q}_{1ref} - \dot{q}_1\right) - \ddot{q}_1 k_{D1} \tag{9}$$

$$v_j = k_{P2}\left(\dot{q}_{2ref} - \dot{q}_2\right) - \ddot{q}_2 k_{D2} \tag{10}$$

being, $\dot{q}_{1ref}$, $\dot{q}_{2ref}$ the reference angular velocities for the PD controller velocities; $k_{P1}$, $k_{P2}$; and $k_{D1}$, $k_{D2}$ are the proportional and derivative gain constants. By ordering the terms of Equations (9) and (10) results in a compact structure for the controller design, which is expressed as follows:

$$\begin{bmatrix} v_i \\ v_j \end{bmatrix} = \begin{bmatrix} k_{P1} & 0 \\ 0 & k_{P2} \end{bmatrix} \begin{bmatrix} \dot{q}_{1ref} \\ \dot{q}_{2ref} \end{bmatrix} - \begin{bmatrix} k_{P1} & 0 \\ 0 & k_{P2} \end{bmatrix} \begin{bmatrix} \dot{q}_1 \\ \dot{q}_2 \end{bmatrix} - \begin{bmatrix} k_{D1} & 0 \\ 0 & k_{D2} \end{bmatrix} \begin{bmatrix} \ddot{q}_1 \\ \ddot{q}_2 \end{bmatrix} \tag{11}$$

$$\mathbf{v_r} = \mathbf{J}\dot{\mathbf{q}}_{\mathbf{ref}} - \mathbf{J}\dot{\mathbf{q}} - \mathbf{P}\ddot{\mathbf{q}} \tag{12}$$

where $\dot{\mathbf{q}}_{\mathbf{ref}}$ is the vector of reference angular velocities, and $\mathbf{J}$, $\mathbf{P}$ are gain matrices.

Associating Equations (8) and (12), is obtained:

$$\boldsymbol{\tau_p} = \mathbf{S}\left[\mathbf{J}\dot{\mathbf{q}}_{\mathbf{ref}} - \mathbf{J}\dot{\mathbf{q}} - \mathbf{P}\ddot{\mathbf{q}}\right] - \mathbf{G}\dot{\mathbf{q}} \tag{13}$$

Relating Equations (4) and (13) gives:

$$\mathbf{S}\left[\mathbf{J}\dot{\mathbf{q}}_{\mathbf{ref}} - \mathbf{J}\dot{\mathbf{q}} - \mathbf{P}\ddot{\mathbf{q}}\right] - \mathbf{G}\dot{\mathbf{q}} = \overline{\mathbf{M}}(\mathbf{q})\ddot{\mathbf{q}} + \overline{\mathbf{C}}(\mathbf{q},\dot{\mathbf{q}})\dot{\mathbf{q}} + \overline{\mathbf{f}}(\mathbf{q}) \tag{14}$$

By ordering Equation (14), a compact dynamic model is obtained, which considers as input signals the velocities of the Scara robot as follows:

$$\dot{\mathbf{q}}_{\mathbf{ref}} = \mathbf{J}^{-1}\left[\mathbf{S}^{-1}\overline{\mathbf{M}}(\mathbf{q}) + \mathbf{P}\right]\ddot{\mathbf{q}} + \mathbf{J}^{-1}\left[\mathbf{S}^{-1}\left[\overline{\mathbf{C}}(\mathbf{q},\dot{\mathbf{q}}) + \mathbf{G}\right] + \mathbf{J}\right]\dot{\mathbf{q}} + \mathbf{J}^{-1}\mathbf{S}^{-1}\overline{\mathbf{f}}(\mathbf{q}) \tag{15}$$

$$\dot{\mathbf{q}}_{\mathbf{ref}}(t) = \mathbf{M}(\mathbf{q})\ddot{\mathbf{q}}(t) + \mathbf{C}(\mathbf{q},\dot{\mathbf{q}})\dot{\mathbf{q}}(t) + \mathbf{f}(\mathbf{q}) \tag{16}$$

It is worth mentioning that for the third joint, being an in-dependent joint, the mathematical model of a DC servomotor obtained from a previous work is considered [27].

## 5. Control Scheme

The closed-loop control scheme proposed in Figure 7 allows to position the operating endpoint in different desired positions, to execute autonomous tasks within the virtual training system. The proposed controller considers a kinematic control and a dynamic compensation control.

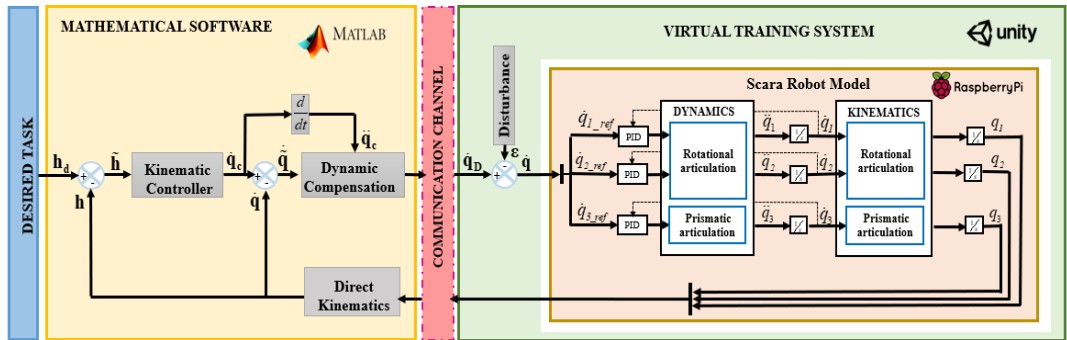

**Figure 7.** Proposed control scheme.

### 5.1. Kinematic Control

The kinematic controller is based on the kinematic model of the Scara robot, determined previously in Equation (3), and is represented as follows:

$$\dot{\mathbf{q}}_{\mathbf{c}} = \mathbf{J}^{-1}\left[\dot{\mathbf{h}}_{\mathbf{d}} + \mathbf{W}\tanh(\alpha(\mathbf{h}_{\mathbf{d}} - \mathbf{h}))\right] \tag{17}$$

where $\dot{\mathbf{q}}_{\mathbf{c}}$ represents the vector of velocities of the robot joints; $\mathbf{J}^{-1} = \text{Adj}^T(\mathbf{J})/|\mathbf{J}|$ with $|\mathbf{J}| \neq 0$ is the inverse matrix of the robotic system kinematics; $\dot{\mathbf{h}}_{\mathbf{d}}$ is the vector of desired velocities; $\tilde{\mathbf{h}}$ represents the control error defined by; $\tilde{\mathbf{h}} = \mathbf{h}_{\mathbf{d}} - \mathbf{h}$; $\mathbf{W}$, is a positive definite gain matrix [28]; the function $\tanh(.)$ is an analytical saturation limiting control error $\mathbf{h}$, with values de $\alpha > 0$.

Stability Analysis

Position control errors $\tilde{\mathbf{h}}$ are analyzed with respect to a velocity tracking under ideal conditions, therefore it is considered that; $\dot{\mathbf{q}}(t) \equiv \dot{\mathbf{q}}_{\mathbf{c}}(t)$; Substituting Equation (17) into Equation (3) is obtains:

$$\dot{\tilde{\mathbf{h}}} + \mathbf{W}\tanh\left(\alpha\tilde{\mathbf{h}}\right) = 0 \tag{18}$$

The following Lyapunov candidate function is considered for stability analysis $\mathbf{V}\left(\tilde{\mathbf{h}}\right) = \frac{1}{2}\tilde{\mathbf{h}}^{\mathbf{T}}\tilde{\mathbf{h}} < 0$; where the first derivative of the function is $\dot{\mathbf{V}}\left(\tilde{\mathbf{h}}\right) = \tilde{\mathbf{h}}^{T}\dot{\tilde{\mathbf{h}}}$. Replacing Equation (18) in the Lyapunov candidate function gives the following results:

$$\dot{\mathbf{V}}\left(\tilde{\mathbf{h}}\right) = -\tilde{\mathbf{h}}^{T}\mathbf{W}\tanh\left(\alpha\tilde{\mathbf{h}}\right) < 0 \tag{19}$$

The stability of the implemented control loop is guaranteed, so that the position error of the operating endpoint $\tilde{\mathbf{h}}(t) \to 0$ is asymptotically stable with $\mathbf{t} \to \infty$ [29].

### 5.2. Dynamic Compensation Controller

Its objective is to compensate the dynamics of the robotic system to minimize velocity errors. Therefore, a control law based on the dynamic model described above in Equation (16) is used as follows:

$$\dot{\mathbf{q}}_{\mathbf{D}} = \mathbf{M}\left[\ddot{\mathbf{q}}_{\mathbf{c}} + \mathbf{K}\tanh\left(\beta\left(\dot{\tilde{\mathbf{q}}} = \dot{\mathbf{q}}_{\mathbf{c}} - \dot{\mathbf{q}}\right)\right)\right] + \mathbf{C}\dot{\mathbf{q}} + \mathbf{f} \tag{20}$$

where $\dot{\mathbf{q}}_{\mathbf{D}}$ represents the vector of dynamic velocities; $\dot{\mathbf{q}}$ is the vector of real velocities of the robotic system; $\ddot{\mathbf{q}}_{\mathbf{c}}$ is the acceleration vector; $\dot{\tilde{\mathbf{q}}}$ is the difference in variation between desired and actual velocities $\dot{\tilde{\mathbf{q}}} = \dot{\mathbf{q}}_{\mathbf{c}} - \dot{\mathbf{q}}$; the function $\tanh\left(\beta\dot{\tilde{\mathbf{q}}}\right)$ allows the saturation of velocity errors; $\mathbf{K}$ is a positive definite gain matrix and the gain constant $\beta > 0$.

### 5.2.1. Stability Analysis

For the stability analysis of the dynamic compensator, the candidate Lyapunov function was used $\mathbf{V}\left(\dot{\tilde{\mathbf{q}}}\right) = \frac{1}{2}\dot{\tilde{\mathbf{q}}}^{\mathbf{T}}\dot{\tilde{\mathbf{q}}} < 0$; where the first derivative of the function is $\dot{\mathbf{V}}\left(\dot{\tilde{\mathbf{q}}}\right) = \dot{\tilde{\mathbf{q}}}^{T}\ddot{\tilde{\mathbf{q}}}$. Replacing Equations (20) and (16) in the Lyapunov candidate function results in:

$$\dot{\mathbf{V}}\left(\dot{\tilde{\mathbf{q}}}\right) = -\dot{\tilde{\mathbf{q}}}^{T}\mathbf{K}\tanh\left(\beta\dot{\tilde{\mathbf{q}}}\right) < 0 \tag{21}$$

The stability of the implemented control loop is guaranteed, so that the velocity error $\dot{\tilde{\mathbf{q}}}(t) \to 0$ is asymptotically stable with $\mathbf{t} \to \infty$ [29].

### 5.2.2. Robustness Analysis

The robustness analysis is focused on the kinematic controller, and the velocity tracking is not considered appropiate $\dot{\mathbf{q}}_{\mathbf{D}}(t) \neq \dot{\mathbf{q}}(t)$ [30]. This velocity error is produced by perturbations, therefore the candidate Lyapunov function is defined as follows; $\mathbf{V}\left(\tilde{\mathbf{h}}\right) = \frac{1}{2}\tilde{\mathbf{h}}^{\mathbf{T}}\tilde{\mathbf{h}} < 0$; where the first derivative of the function is $\dot{\mathbf{V}}\left(\tilde{\mathbf{h}}\right) = \tilde{\mathbf{h}}^{T}\dot{\tilde{\mathbf{h}}}$. Subsequently, the following is considered $\dot{\mathbf{q}} = \dot{\mathbf{q}}_{\mathbf{D}} + \varepsilon$, where $\varepsilon$ represents the disturbances that are generated at the moment of entering velocity variations in the different actuators (DC motors) that allow the movement of the two links of the robotic system.

Replacing Equation (17) in (3) results in:

$$\dot{\tilde{\mathbf{h}}} = -\mathbf{W}\tanh\left(\alpha\tilde{\mathbf{h}}\right) - \mathbf{J}\varepsilon \tag{22}$$

Substituting Equation (22) in the first derivative of the candidate Lyapunov function gives:

$$\dot{\mathbf{V}}\left(\tilde{\mathbf{h}}\right) = -\tilde{\mathbf{h}}^{T}\mathbf{W}\tanh\left(\alpha\tilde{\mathbf{h}}\right) - \dot{\tilde{\mathbf{h}}}^{T}\mathbf{J}\varepsilon \tag{23}$$

For to it to be negative, it has to be $\dot{\mathbf{V}}\left(\tilde{\mathbf{h}}\right) < 0$:

$$\left|\tilde{\mathbf{h}}^{\mathbf{T}}\mathbf{W}\tanh\left(\alpha\tilde{\mathbf{h}}\right)\right| > \left|\dot{\tilde{\mathbf{h}}}^{T}\mathbf{J}\varepsilon\right| \tag{24}$$

Therefore, for large values of $\tilde{\mathbf{h}}$, it is considered that $\mathbf{W}\tanh\left(\alpha\tilde{\mathbf{h}}\right) \approx \mathbf{W}$, as shown.

$$\|\mathbf{W}\| > \|\tilde{\mathbf{h}}^{\mathbf{T}}\mathbf{J}\boldsymbol{\varepsilon}\| \tag{25}$$

Thus, reducing errors. Whereas, for small values of $\tilde{\mathbf{h}}$ is considered that $\mathbf{W}\tanh\left(\alpha\tilde{\mathbf{h}}\right) \approx \mathbf{W}\tilde{\mathbf{h}}$, therefore:

$$\|\tilde{\mathbf{h}}\| > \frac{\|\mathbf{J}\boldsymbol{\varepsilon}\|}{\lambda_{\min}(\mathbf{W})} \tag{26}$$

Therefore, it can be concluded that control error $\tilde{\mathbf{h}}$ is finally bounded by:

$$\|\tilde{\mathbf{h}}(t)\| \leq \frac{\|\mathbf{J}\boldsymbol{\varepsilon}\|}{\lambda_{\min}(\mathbf{W})} \tag{27}$$

## 6. Analysis and Results

This section presents the results obtained from the virtual training system by applying two simulation techniques: Full simulation and the Hard-ware-in-the-Loop (HIL) technique. It consists of a laboratory environment and an industrial environment developed in Unity 3D software. All this with the aim of improving the teaching and learning process in industrial robotics area. This section is divided into four parts. (i) Operability Interface, where the user can select the hierarchy of operation that he/she will have in the chosen environments. (ii) Virtual Simulator of the Laboratory Environment, where interactive windows are presented that allow the configuration of the VR environment to generate the movements of the robotic system. To obtain the results of the implementation of the advanced control algorithms when performing position and trajectory tasks. (iii) Virtual simulator of the industrial environment, where the automation of the bottle filling process executed by two robotic systems is presented. (iv) Usability of the virtual training system, to determine whether the virtual system meets all the expectations of operation raised, a usability test was performed on a group of 25 people who experimented with the developed environments.

### 6.1. Operability Interface

For the design of the virtual training system interface, the ISO 25010 standard was used as a reference, which deals specifically with the standardization and quality of a software product in compliance with established standards and parameters [31].

Figure 8 shows an interactive main window, which allows the user to select the hierarchy of operation, in the chosen environment, as follows: (i) Operator, will assign tasks within the environments. (ii) Supervisor, is the highest level and allows to modify all the parameters that are in the scenarios, in addition to generating disturbances. (iii) Guest, spectator with certain restrictions on the tasks performed by the operator and the supervisor.

### 6.2. Laboratory Environment

The environment represents a simple and easy to operate structure as shown in Figure 9, where tests are performed using the Full simulation technique to verify the behavior of the robot, to execute position or trajectory tasks. When working in operator mode, certain options are enabled and can be configured by the user in the control window, such as: assign position and trajectory tasks. The supervisor mode can perform all activities that the operator is responsible for. In addition, it allows it to establish disturbances. In Guest mode, the user visualizes the assigned tasks; together, a window is added where all the results obtained from the implementation of the advanced control algorithm are presented.

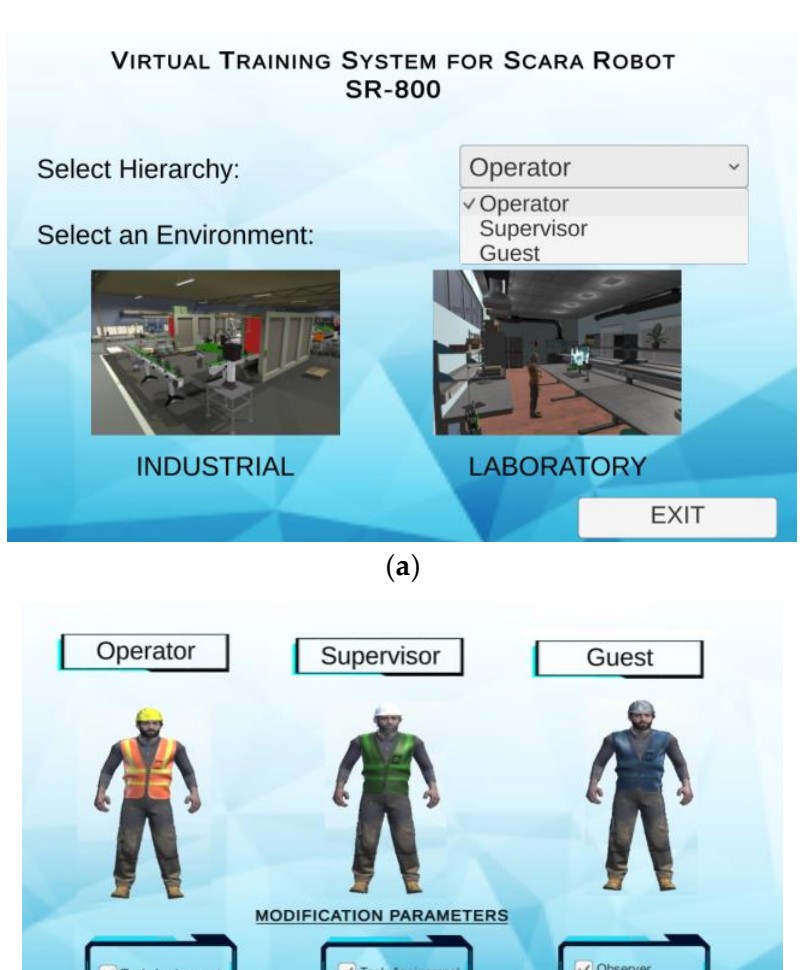

(**a**)

(**b**)

**Figure 8.** Operability interface for virtual laboratory and industrial training system.(**a**) Operating interface of the virtual training system. (**b**) User selection interface according to hierarchy level.

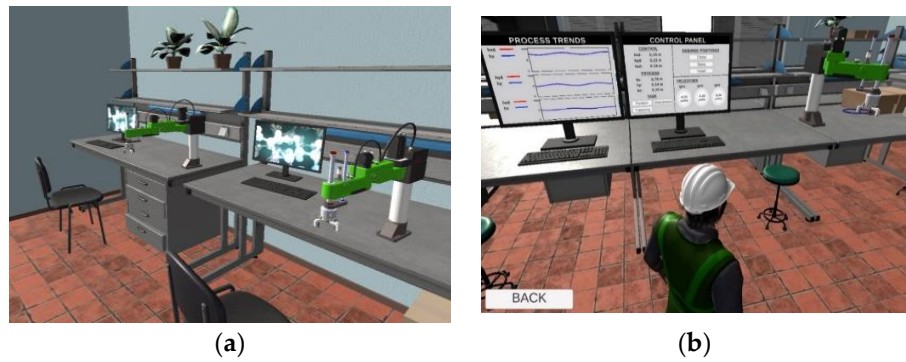

(**a**)　　　　　　　　　　　　　　　　(**b**)

**Figure 9.** Virtualized laboratory environment for advanced controller application and testing of assigned tasks.(**a**) Laboratory Environment (**b**) Assigned tasks.

*6.3. Implementation of the Control Algorithm*

For the execution of the circular trajectory in the laboratory environment, the trajectory control of a Scara robot was considered. Table 1 shows the most relevant parameters

applied to the proposed control scheme as follows: Initial conditions of the robot joints $\mathbf{q}(q_1, q_2, q_3)$; the weight matrices of control errors $\mathbf{W}$ and $\mathbf{K}$; the gain constants $\alpha$ and $\beta$. All this in order to execute the desired trajectory $\mathbf{h_d} = \left[h_{xd}, h_{yd}, h_{zd}\right] \in \mathrm{R}^3$. In addition, for the tests carried out, a sampling time of $T_0 = 0.1[s]$.

**Table 1.** Desired task and initial parameters.

| Initial Condition q(0) | | Gain Parameters | | Desired Task h$_\mathbf{d}$(t) | |
|---|---|---|---|---|---|
| $q_1(0)$ | 0.1745[rad] | $\mathbf{W} \in R^{3\times3}$ | $diag(2, 2, 1.5)$ | $h_{xd}(t)$ | $0.5 + 0.2\sin(0.08t)$[m] |
| $q_2(0)$ | 0.5235[rad] | $\alpha \in R^+$ | 0.85 | $h_{yd}(t)$ | $0.2\cos(0.08t)$[m] |
| $q_3(0)$ | 0[rad] | $\mathbf{K} \in R^{3\times3}$ | $diag(1.5, 1.5, 1)$ | $h_{zd}(t)$ | $0.5 + 0.2\sin(0.08t)$[m] |
| | | $\beta \in R^+$ | 0.75 | | |

Figure 10 presents the results obtained from the behavior of the Robot in the laboratory environment, when implementing the control scheme proposed in Section 5. Figure 10a shows the behavior of the actual robot trajectory $\mathbf{h}(h_x, h_y, h_z)$ with respect to the desired trajectory $\mathbf{h_d}\left(h_{xd}, h_{yd}, h_{zd}\right)$. The desired trajectory $\mathbf{h_d}$, corresponds to a trigonometric circle to be executed by the Robot. The control error of the operating end is defined as the difference between the desired and the actual path, i.e., $\tilde{\mathbf{h}} = \mathbf{h_d} - \mathbf{h}$; therefore, these errors converge to values close to zero asymptotically, i.e., reaching final characteristic errors max $\|\tilde{\mathbf{h}}(t)\| < 0.01$ [m], as shown in Figure 10b. Finally, Figure 10c shows the velocity control errors $\dot{\tilde{\mathbf{q}}}(t) = \dot{\mathbf{q}}_\mathbf{c}(t) - \dot{\mathbf{q}}(t)$, is defined by the difference between the velocity obtained from the kinematic controller and the velocity of the robot. In this test, the limit of the maximum velocity error is max $\|\dot{\tilde{\mathbf{q}}}(t)\| < 0.02$ [rad/s].

*6.4. Industrial Environment*

Industrial processes are related to the interaction between humans and machines. By implementing virtual reality, manufacturing simulations are created with the objective of presenting a high degree of realism and optimization in the process line [32,33]. In this industrial environment the HIL simulation technique is applied, working in operator mode the user has the option to activate and deactivate the automated bottle filling process from the control panel. The supervisor mode performs the same tasks as the operator and has access to modify velocities to generate disturbances. In guest mode this will be part of a spectator without any interaction with the process. On the other hand, two windows are presented with the results obtained from the behavior of the robotic system when the advanced control algorithm was implemented.

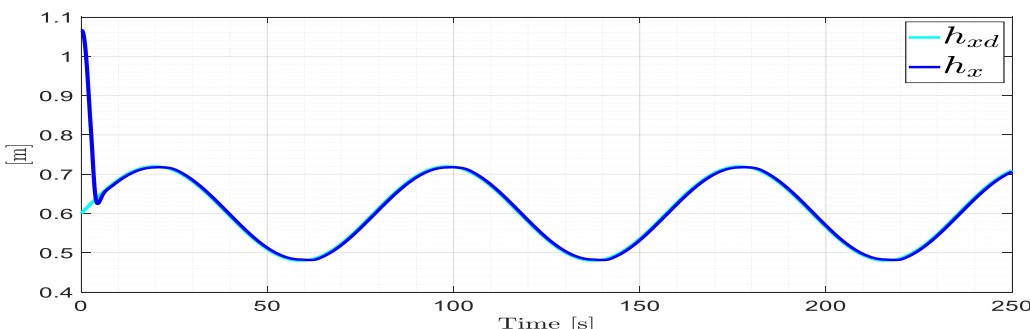

**Figure 10.** *Cont.*



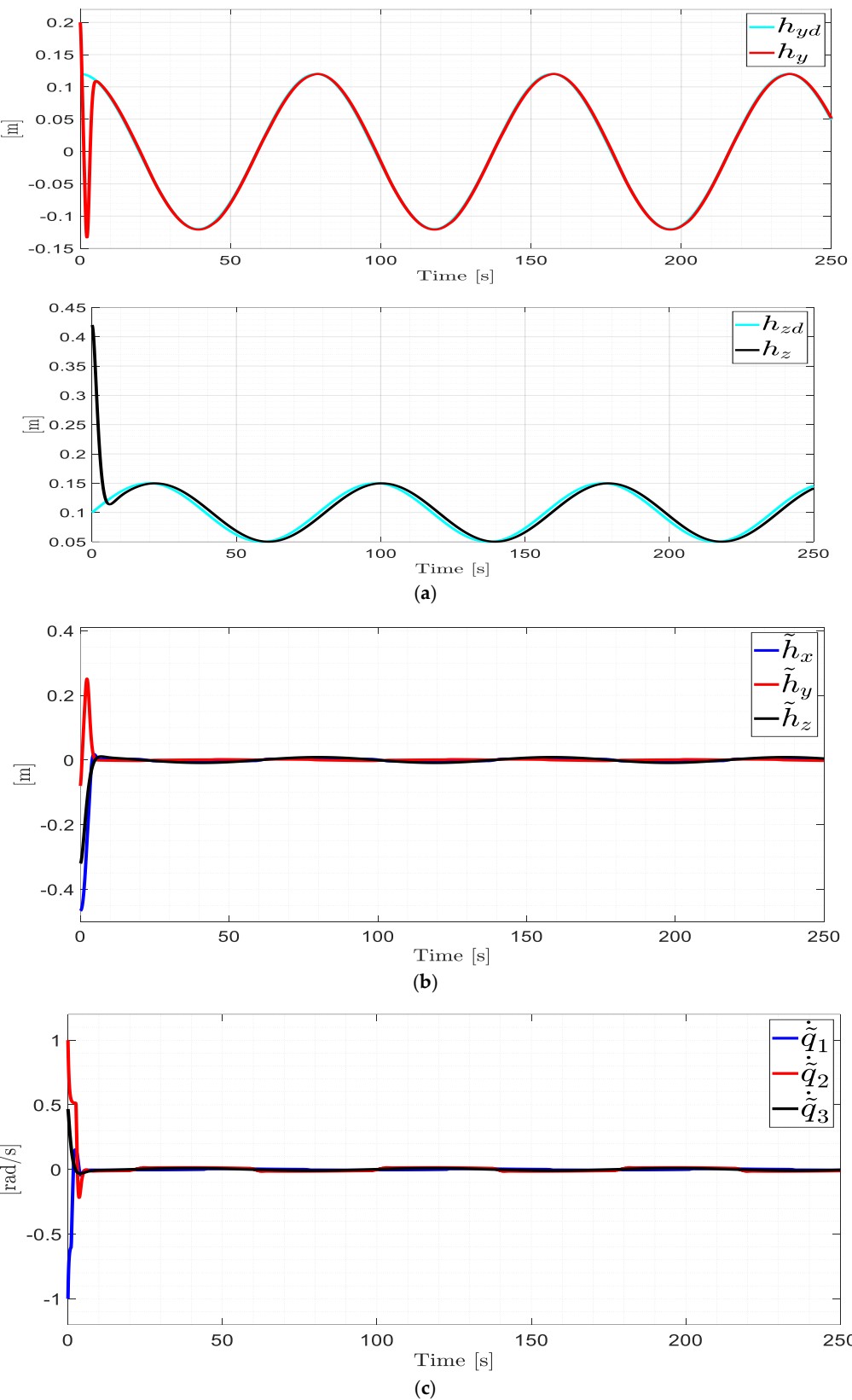

**Figure 10.** Movements of the Scara robot while executing a trajectory task in the laboratory environment using the Full simulation technique. (**a**) Desired trajectory and actual trajectory executed by the SCARA robot. (**b**) Robot operating end control errors. (**c**) Velocity control errors.

The process that takes place in the Unity environment, as shown in Figure 11, describes the steps as follows: Figure 11a Disinfection and transfer of bottles, which receive an automatic purification and drying treatment. Figure 11b Box assembly and transfer, at this stage the boxes are assembled and sent to the packing process. Figure 11c Fault detector, using a vision sensor installed on the conveyor belt, scans the containers for defects or anomalies. Figure 11d Filling, capping, and labeling, the first robot manipulator of Scara SR-800 configuration named as A, receives the orders from the sensor and moves the empty bottles to continue with the process line. Figure 11e Packaging, the second robotic arm designated as B, receives the signal from a proximity sensor which allows it to detect and arrange the bottles in 6 different positions in a packing boxes, using an advanced controller for positional tasks. Figure 11f Sealing, the boxes with the product are sealed and stored, ending the process.

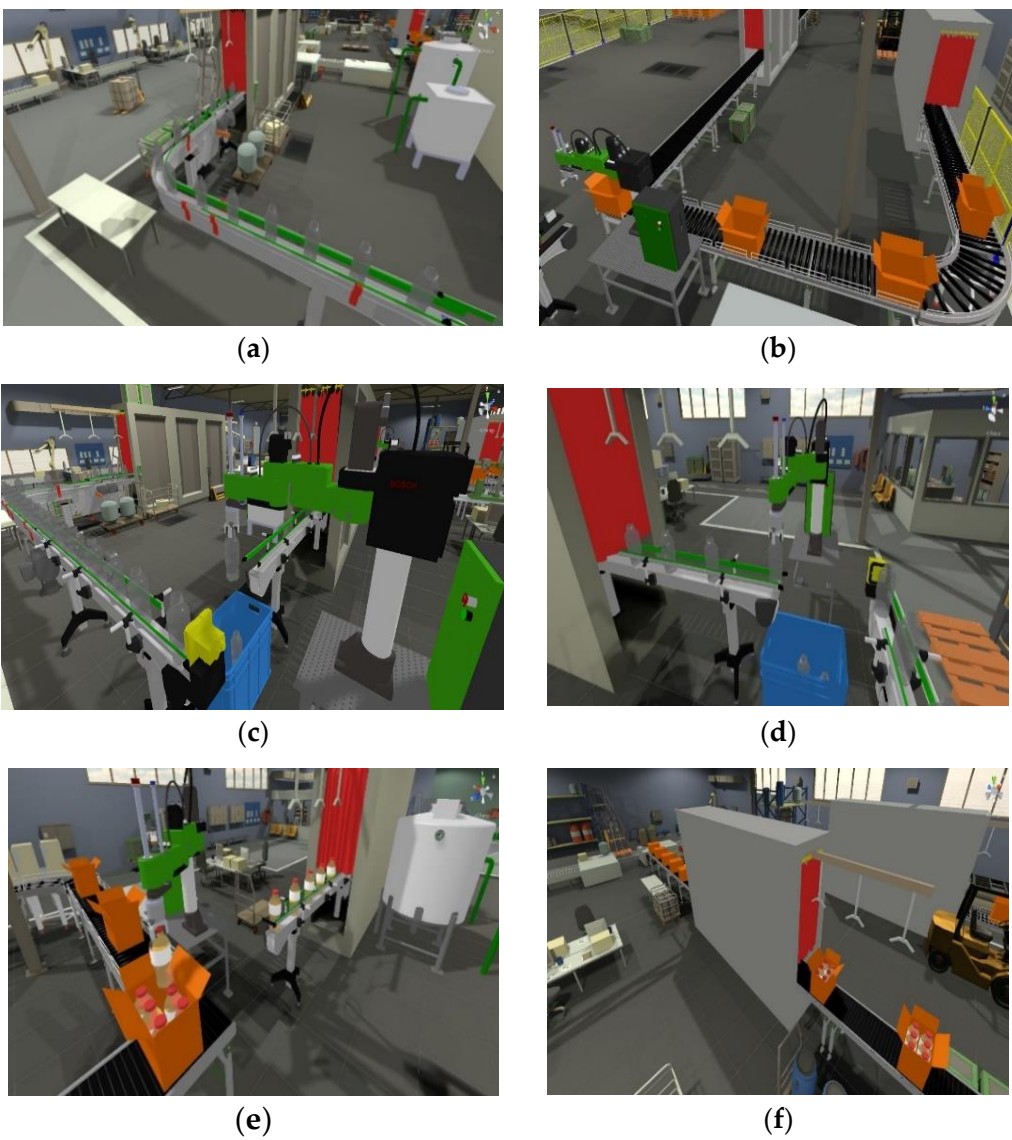

**Figure 11.** Description of the stages of the industrial environment applied to the automatic bottle filling process using robotic systems. (**a**) Disinfecting and transfer of bottles. (**b**) Assembly and transfer of boxes. (**c**) Failure detector for bottles. (**d**) Filling, capping and labeling. (**e**) Bottle packaging. (**f**) Sealing boxes.

### 6.5. Implementation of the Control Algorithm

For the execution of the bottle packaging process, the positioning control of two Scara robots, robot A, robot B, was considered. Table 2 represents the parameters considered for the implementation of the control scheme i.e.,: initial conditions of the robot joints $\mathbf{q_A}(q_1, q_2, q_3)$, $\mathbf{q_B}(q_1, q_2, q_3)$; control error weight matrices $\mathbf{W}$ and $\mathbf{K}$; gain constants $\alpha$ and $\beta$. The desired positions of the operating end of each robot are defined as: $\mathbf{h_{dA}} = \begin{bmatrix} h_{xd}, h_{yd}, h_{zd} \end{bmatrix} \in \mathrm{R}^3$ and $\mathbf{h_{dB}} = \begin{bmatrix} h_{xd}, h_{yd}, h_{zd} \end{bmatrix} \in \mathrm{R}^3$. It should be emphasized that the position control strategy [30] was considered for both Robot A and Robot B in order to execute the bottle packaging process. For the position control, it is considered that the desired positions of the robots are constant, therefore, the desired velocities are equal to zero, i.e., $\mathbf{h_{dA}}$ = constant, $\mathbf{h_{dB}}$ = constant; and $\dot{\mathbf{h}}_{\mathbf{dA}} = \dot{\mathbf{h}}_{\mathbf{dB}} = \mathbf{0} \in R^3$. Finally, the tests were performed with a sampling time of $T_0 = 0.1[s]$.

**Table 2.** Desired task and initial parameters.

| Initial Condition q(0) | | Gain Parameters | | * Desired Task h$_{dA}$, h$_{dB}$ | |
|---|---|---|---|---|---|
| **Robot A** | | | | | |
| $q_1(0)$ | 0.1745[rad] | $\mathbf{W} \in R^{3\times3}$ | $diag(2,\ 2,\ 1.5)$ | $h_{xd}(t)$ | $f_x(\rho, \wedge)[\mathrm{m}]$ |
| $q_2(0)$ | 0.5235[rad] | $\alpha \in R^+$ | 0.85 | $h_{yd}(t)$ | $f_y(\rho, \wedge)[\mathrm{m}]$ |
| $q_3(0)$ | 0[rad] | $\mathbf{K} \in R^{3x3}$ | $diag(1.5,\ 1.5,\ 1)$ | $h_{zd}(t)$ | $f_z(\rho, \wedge)[\mathrm{m}]$ |
| | | $\beta \in R^+$ | 0.75 | | |
| **Robot B** | | | | | |
| $q_1(0)$ | 0.3490[rad] | $\mathbf{W} \in R^{3\times3}$ | $diag(2,\ 1.5,\ 1.5)$ | $h_{xd}(t)$ | $f_x(\rho, \wedge)[\mathrm{m}]$ |
| $q_2(0)$ | 0.5235[rad] | $\alpha \in R^+$ | 0.95 | $h_{yd}(t)$ | $f_y(\rho, \wedge)[\mathrm{m}]$ |
| $q_3(0)$ | 0[rad] | $\mathbf{K} \in R^{3x3}$ | $diag(1.9,\ 1.9,\ 1)$ | $h_{zd}(t)$ | $f_z(\rho, \wedge)[\mathrm{m}]$ |
| | | $\beta \in R^+$ | 0.80 | | |

\* $f(\rho, \wedge)$ represents a set of desired positions of the operating end of each robot.

Figure 12 presents the results obtained from the behavior of Robot A when implementing the control scheme proposed in Section 5. Figure 12a shows the behavior of the real robot positions of the robot $\mathbf{h_A}(h_x, h_y, h_z)$ with respect to the desired positions $\mathbf{h_{dA}}\left(h_{xd}, h_{yd}, h_{zd}\right)$. Desired positions $\mathbf{h_{dA}}$, are defined as a set of desired points as a function of the bottle selection task to be executed by Robot A. The first graph in Figure 12a shows the actual position of the robot $(h_x)$ on the X-axis, which differs from the desired position $h_{xd}$, if and only if the desired value $\left(h_{yd}\right)$ on the Y-axis position is modified. However, the implemented control algorithm makes the error $\widetilde{h}_x = h_{xd} - h_{xd}$ on the X-axis tend back to zero asymptotically. On the other hand, the control error is defined between the difference of the desired and real position i.e., $\widetilde{\mathbf{h}}_{\mathbf{A}} = \mathbf{h_{dA}} - \mathbf{h_A}$; therefore, in this paper the change of the desired position is considered, when $\rho_A < 0.001$ [m], defined as $\rho_A(t) < \|\widetilde{\mathbf{h}}_{\mathbf{A}}(t)\|^2$, as shown in Figure 12b. Figure 12c shows the velocities calculated by the kinematic controller Equation (17). Finally, Figure 12d presents the maneuverability velocities calculated by the dynamic compensation controller Equation (20) so that the control error tends to zero when the time tends to infinity, that is, asymptotic stability. It is worth mentioning that the maneuverability velocities are limited between $\left|\dot{q}_{Di}\right| \le 2$ [rad/s], with $i = 1, 2, 3$ $i = 1, 2, 3$.

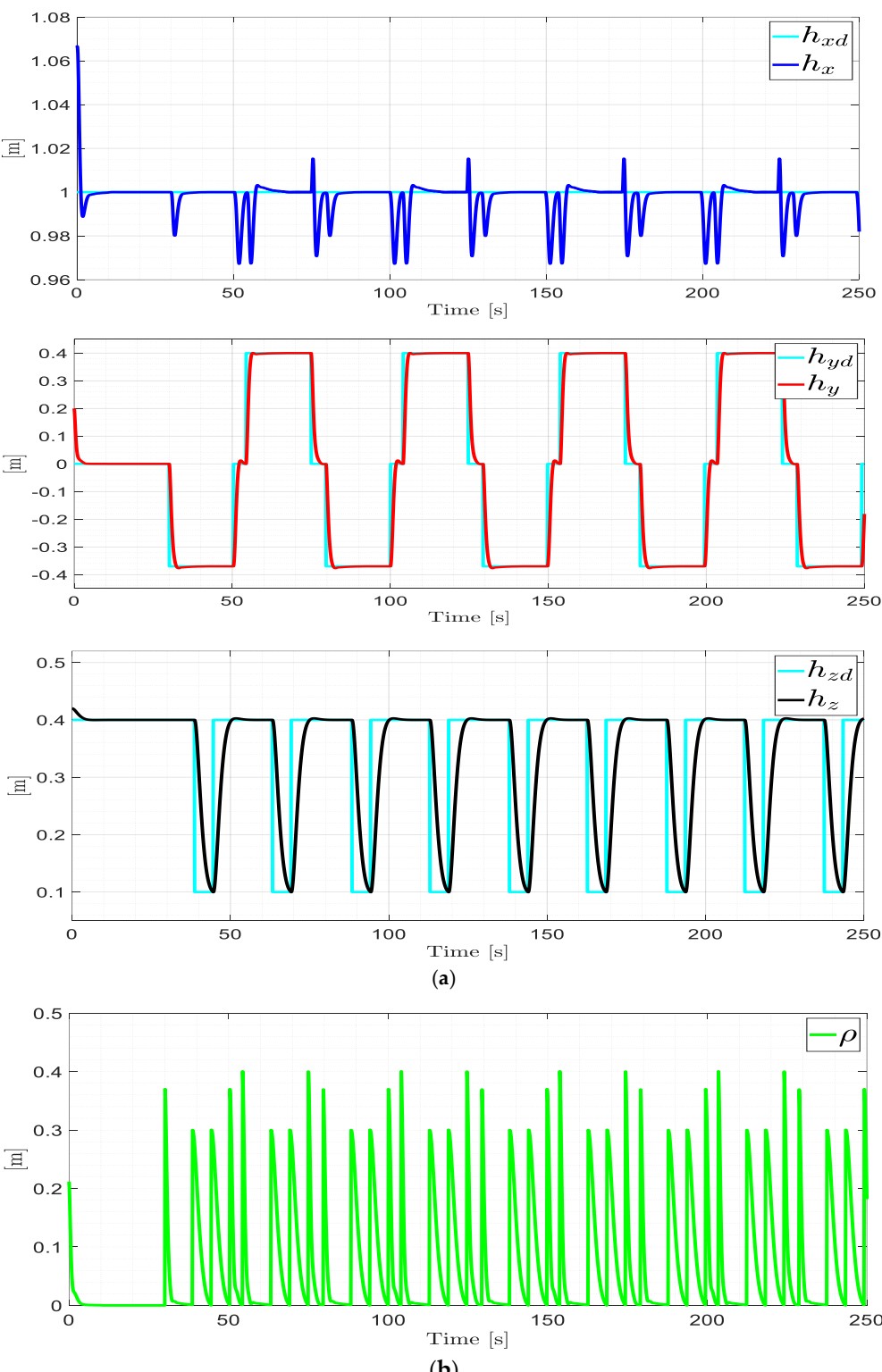

**Figure 12.** *Cont.*

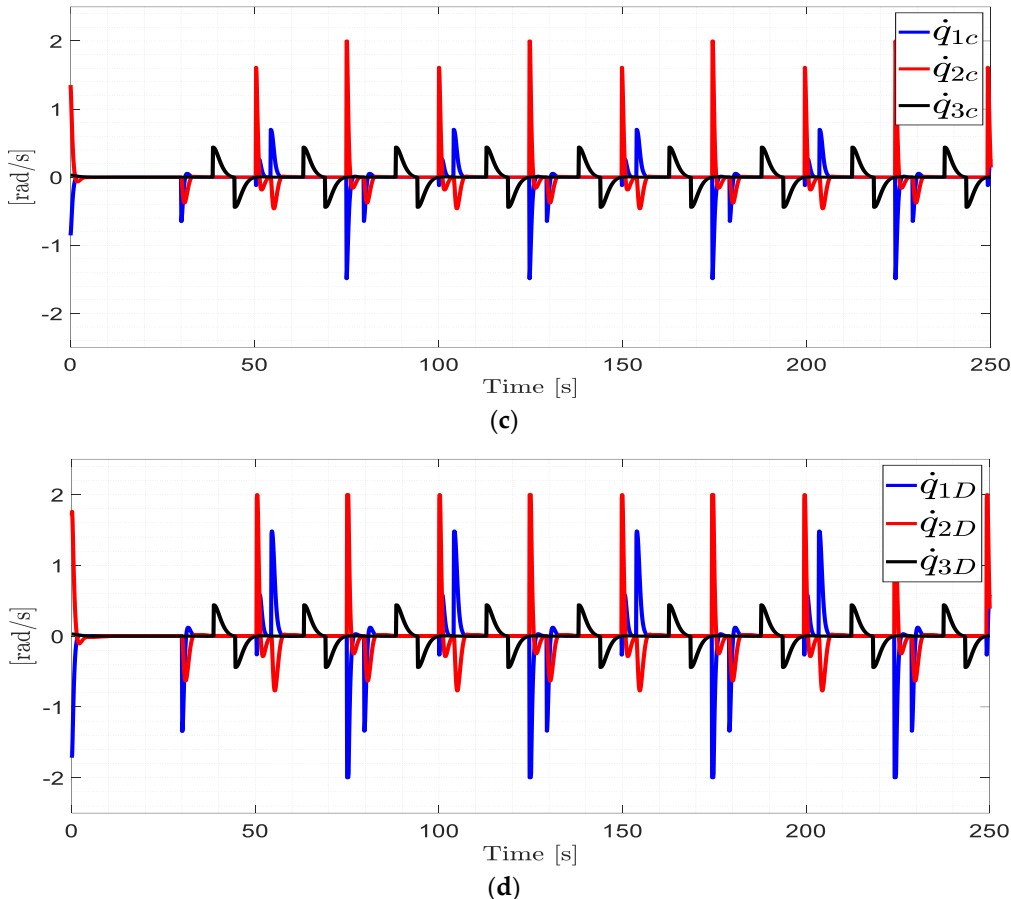

**Figure 12.** Movements of the Scara A robot in the virtualized industrial environment, when performing the bottle selection task; (**a**) Desired and real robot operating end positions; (**b**) Distance between the desired positions and the real position of the robot $\rho_A(t) < \left\| \tilde{\mathbf{h}}_\mathbf{A}(t) \right\|^2$; (**c**) Velocities calculated by the kinematic controller; (**d**) Maneuverability velocities calculated by the dynamic compensator.

Figure 13 similarly presents the results obtained from the behavior of Robot B when implementing the control scheme of Section 5. Figure 13a shows the real positions $\mathbf{h_B}(h_x, h_y, h_z)$ with respect to the desired positions $\mathbf{h_{dB}}\left(h_{xd}, h_{yd}, h_{zd}\right)$. The desired positions of Robot B are the function of arranging the bottles in their respective packing cases; Figure 13b indicates the distance between the desired positions and the actual position of the robot. The desired change of position is considered, when $\rho_B < 0.001$ [m]. Finally, Figure 13c,d, respectively, show the velocities calculated by the kinematic controller. $(\dot{q}_{1c}, \dot{q}_{2c}, \dot{q}_{3c})$ and by dynamic compensator $(\dot{q}_{1D}, \dot{q}_{2D}, \dot{q}_{3D})$ in order for the control error to tend to zero. Furthermore, the maneuverability velocities of Robot B are limited by $\left| \dot{q}_{Di} \right| \leq 2$ [rad/s], with $i = 1, 2, 3$.

*6.6. Usability of the Virtual Training System*

ISO 9241-11 suggests that usability measures should cover effectiveness, efficiency and user satisfaction in using a system [34]. System Usability (SUS) is a Likert scale based on 10 forced-choice questions to indicate the degree of agreement or disagreement with a statement. To calculate the SUS score, the contributions of each question are summed. The contribution of each question will range between 0–4. For questions 1, 3, 5, 7 and 9, the score is the scale position minus 1. For items 2, 4, 6, 8, and 10, the score is 5 minus the scale position. Multiply the total score by 2.5 to generate the total usability value, the range varies from 0–100 [35]. Finally, for the present work, the usability (SUS) was determined and applied to a group of 25 people with basic knowledge in the robotics

area. Each participant was given the executable application of the virtual environment for its subsequent installation. Following this, all participants were trained in the correct manipulation of the virtual training system by applying simulation techniques such as: Full simulation and HIL. In order to execute position and trajectory tasks.

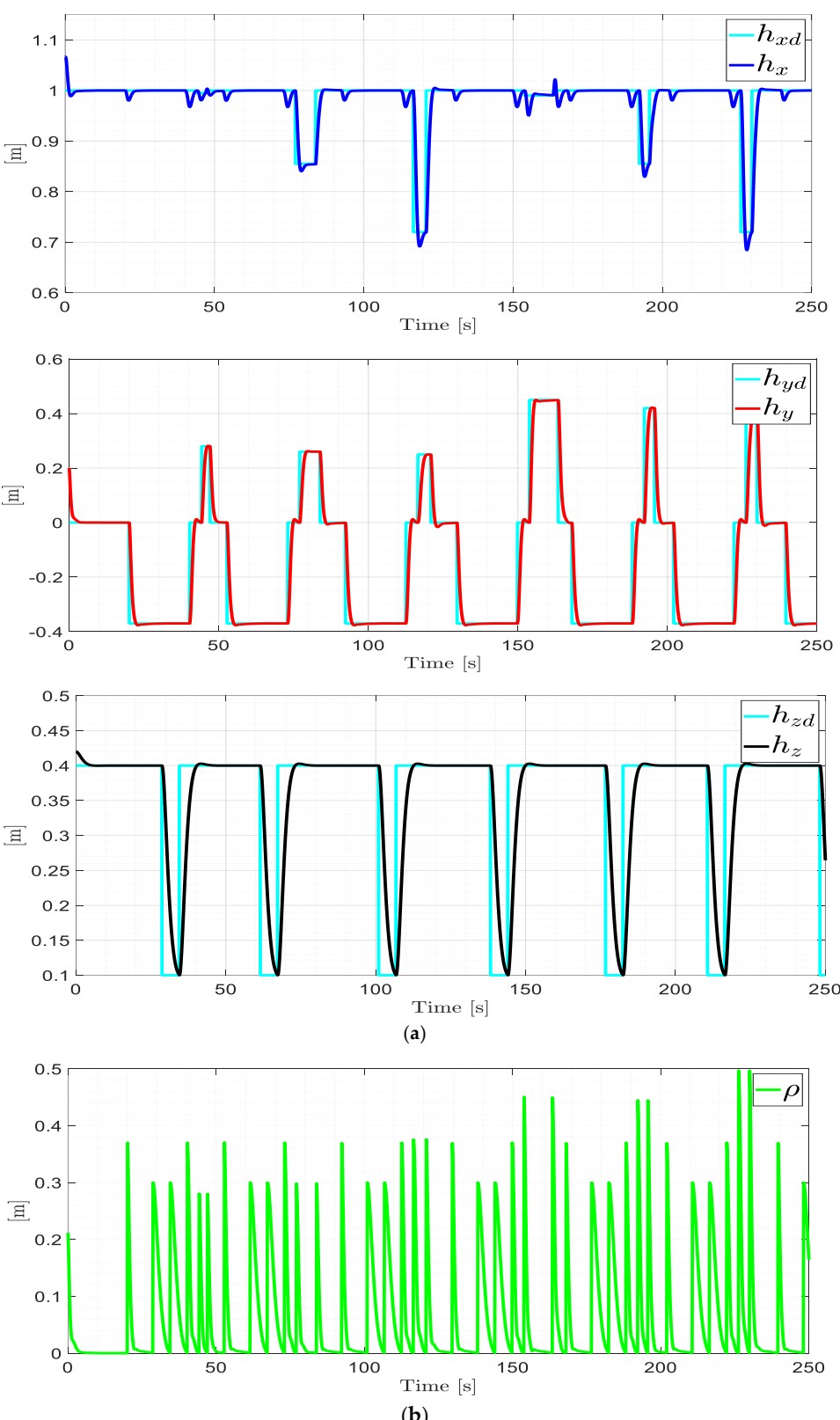

**Figure 13.** *Cont.*

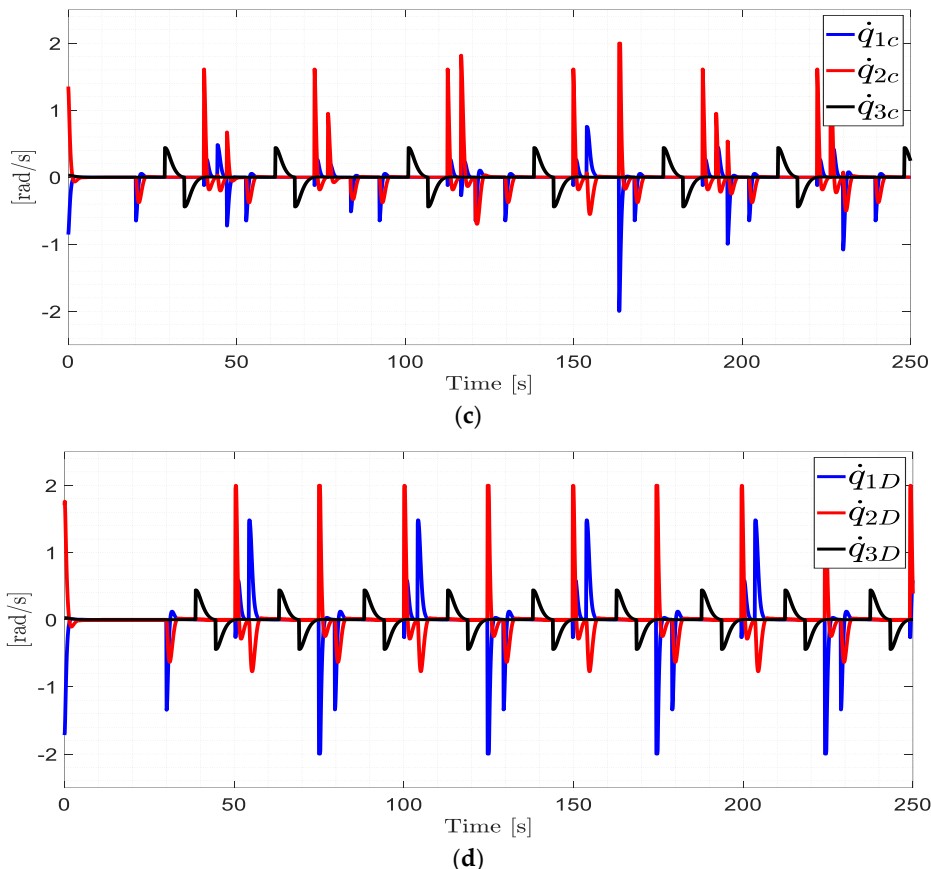

**Figure 13.** Movements of the Scara B robot in the virtualized industrial environment, when performing the bottle selection task; (**a**) Desired and real robot operating end positions; (**b**) Distance between the desired positions and the real position of the robot $\rho_B(t) < \|\tilde{\mathbf{h}}_B(t)\|^2$; (**c**) Kinematic controller velocities; (**d**) Maneuverability velocities calculated by the dynamic compensator.

At the end, a usability score of 87.5 points of acceptance was obtained as shown in Table 3, demonstrating that the environments are very reliable and versatile both in handling and in the interpretation of results, which indicates that it is suitable for teaching and learning [22].

**Table 3.** Usability Testing (SUS).

| No. | Questions | Score | Operation |
|:---:|:---|:---:|:---:|
| 1 | I think I would like to use this virtual training system frequently. | 4 | $4 - 1 = 3$ |
| 2 | I found the training system unnecessarily complex. | 1 | $5 - 1 = 4$ |
| 3 | I thought the training system was easy to use. | 4 | $4 - 1 = 3$ |
| 4 | I think I would need the support of a specialist to be able to use this system. | 1 | $5 - 1 = 4$ |
| 5 | I found that the various functions of this system were well integrated. | 5 | $5 - 1 = 4$ |
| 6 | I thought there was too much inconsistency in this virtual training system. | 2 | $5 - 2 = 3$ |
| 7 | I imagine most people would learn to use this system very quickly. | 4 | $4 - 1 = 3$ |
| 8 | I found the virtual training system very cumbersome to use. | 1 | $5 - 1 = 4$ |
| 9 | I felt very comfortable using the virtual training system. | 5 | $5 - 1 = 4$ |
| 10 | I needed to learn a lot of things before I can use this virtual training system. | 4 | $4 - 1 = 3$ |
| | | **Total** | $34 \times 2.5 = 87.5$ |

## 7. Conclusions

With the application of Full Simulation and HIL simulation techniques to a virtual training system consisting of a laboratory environment and an industrial environment.

The simulation of a robotic arm is tested, executing position and trajectory tasks. Thus improving teaching and learning in the area of industrial robotics. The kinematic and dynamic mathematical models obtained resemble the real behavior of the Scara SR-800 robot. In addition, they are applied for the development of the control algorithm. The implemented control scheme significantly reduces the position and velocity errors of the robotic system, which means that the proposed controller is stable and robust, for both virtualized environments.

**Author Contributions:** Conceptualization V.H.A., J.S.I. and E.J.A.; methodology C.A.N., J.S.I. and E.J.A.; software V.H.A., J.S.I. and E.J.A.; formal analysis C.A.N. and V.H.A.; investigation V.H.A., J.S.I. and E.J.A.; resources J.S.I. and E.J.A.; data curation J.S.I. and E.J.A.; writing—original draft preparation V.H.A., J.S.I. and E.J.A.; writing—review and editing V.H.A., J.S.I., E.J.A. and C.A.N.; visualization V.H.A., J.S.I. and E.J.A.; supervision V.H.A. and C.A.N.; project administration V.H.A.; funding acquisition V.H.A., J.S.I. and E.J.A. All authors have read and agreed to the published version of the manuscript.

**Funding:** Funded by Universidad de las Fuerzas Armadas ESPE.

**Acknowledgments:** The authors would like to thank the Universidad de las Fuerzas Armadas ESPE and to the ARSI Research Group for their support in developing this work.

**Conflicts of Interest:** The authors declare no conflict of interest.

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
