# Peer review of "Virtual Training System for the Teaching-Learning Process in the Area of Industrial Robotics"

_electronics, doi:10.3390/electronics12040974_

Round 1

Reviewer 1 Report

1. A hard space should be inserted between the text and literature reference number (eg. 29).  Comment applies to the entire text of the article.

2. Remove the comma before citing a reference on lines 29 and 41.

3. The editing error on line 14 - Original text “Raspberry pi.". Corrected text “Raspberry Pi..".

4. The editing error on line 302- Subsection title 5.2.2. should be written in English.

5. Authors should add more textual description to the content of the article regarding the results presented in Fig. 11, 12. A more detailed analysis of the results achieved is required.

Reviewer 2 Report

  The paper describes a virtual model or training operation of a robotic arm.  The virtual model simulates the behavior of the real robotic arm, although no experiments with a physical robot were performed.
  In general the paper is clear and well organised, despite the poor quality of English.  Here are some more details to hopefully help improve the paper.

  - Some claims of the paper are not correctly referenced. For example, the source of the numbers presented in the third paragraph of Introduction could be referenced.

  - The 5th paragraph of the Introduction is also very odd. It seems to imply that VR was invented just in the last 10 years, which obviously is not the case. Also, the distinction between paid and open source software is inaccurate: open source software can also be paid, so it would be more accurate to distinguish between closed and open source software.  And more references could be presented.

  - The paper describes that the Matlab control algorithm communicates with the simulator via shared memories. The architectures of those "shared memories," synchronization methods and other details should be described in the paper.

  - Descriptions of the modeling process and figures 1 and 2 are very confuse.  It is hard to understand what is the general concept and what was actually implemented.  I suggest at least Figure 2 is reorganised or split into logical and physical models.  It is important to clarify which models are physical, which ones are virtual, the logical components and interactions, even if that is split between different diagrams.

  - The same as above could also be said about Figure 3, which also mixes logical and physical aspects and becomes difficult to understand.

  - The function of the wireless communication module in a totally simulated environment could also be clarified.

  - Equation 9 is actually two equations.

  - The title of Section 5.2.2 must be translated.

  - Expressions such as "you will have" should be avoided in scientific papers. They occur specially in Section 6.

  - The letter size in the charts is very small and it is only possible to read with a very large zoom.

  - Section 6.5 describes usability tests and acceptance of 85.5%. Details of the tests, statistical validity, how the survey was performed and how the acceptance rate was calculated should be given.
    The text should also be revised, the title of the paper there does not make sense.

  - The most concerning point, however, is that there is no validation of the system compared to the real SCARA SR-800. So the advantages and limitations of the approach could be discussed, noting that all the system is just a simulator inspired by an actually existing robot. The title of the paper should also be more specific.  

  - There are many sentences with incorrect grammatical structure. The paper needs to be revised by someone with good command of English.

Reviewer 3 Report

In the desire to make arguments for the publication of the paper in MDPI Electronics journal, the authors mention that they propose an application in the domain opf “Hardware-in-the-Loop technique in virtual environments for learning and teaching industrial processes”, and that they use a RaspberyPi electronic board.

In fact paper deals with Virtual Training, Modeling and Kinematic Control for Virtual Robotics Application.

Without diminishing the merit of the work, we cannot neglect the fact that we have not identified clear contributions related to electronics nor hardware.

MDPI journals like Automation, Modelling or Robotics would be more appropriate to the actual content of the work.

Reviewer 4 Report

Please proceed to the attached file. 

Round 2

Reviewer 2 Report

The paper has been greatly improved and my main concerns have been well addressed.

Reviewer 3 Report

I appreciate the determination of the authors to publish in this journal and not in another.

I would venture to say that there are approaches to "Extended Reality (XR): AR, VR, MR and Beyond", even with a stronger connection to what engineers identify as being electronics